# INDUCING DYSLEXIA IN VISION LANGUAGE MODELS

**Melika Honarmand, Ayati Sharma, Badr AlKhamissi, Johannes Mehrer**[*]**& Martin Schrimpf**[*]

École Polytechnique Fédérale de Lausanne (EPFL)
Lausanne, Switzerland
`{melika.honarmand, badr.alkhamissi}@epfl.ch`
`{johannes.mehrer, martin.schrimpf}@epfl.ch`
`ayati.sharma@berkeley.edu`

## ABSTRACT

Dyslexia, a neurodevelopmental disorder characterized by persistent reading difficulties, is often linked to reduced activity of the visual word form area (VWFA) in the ventral occipito-temporal cortex. Traditional approaches to studying dyslexia, such as behavioral and neuroimaging methods, have provided valuable insights but remain limited in their ability to test causal hypotheses about the underlying mechanisms of reading impairments. In this study, we use large-scale vision-language models (VLMs) to simulate dyslexia by functionally identifying and perturbing artificial analogues of word processing. Using stimuli from cognitive neuroscience, we identify visual-word-form-selective units within VLMs and demonstrate that they predict human VWFA neural responses. Ablating model VWF units leads to selective impairments in reading tasks while general visual and language comprehension abilities remain intact. In particular, the resulting model matches dyslexic humans' phonological deficits without a significant change in orthographic processing, and mirrors dyslexic behavior in font sensitivity. Taken together, our modeling results replicate key characteristics of dyslexia and establish a computational framework for investigating brain disorders. [1]

## 1 INTRODUCTION

Dyslexia is a complex neurodevelopmental learning disorder that impairs a person's ability to decode written language and to spell, despite normal intelligence and educational opportunity. Difficulties arise mainly in reading, spelling, accuracy, fluency, and decoding abilities (Snowling et al., 2020; Kunwar & Sapkota, 2022). Depending on the severity criteria used, dyslexia is estimated to impact 6–17% of children in the school-age population (Pennington et al., 2009; Lyon et al., 2003), with the global prevalence reported to range from less than 5% to 20% of the entire population (Wagner et al., 2020). The brain disorder is associated with atypical neural activity in regions involved in language processing and phonological representation, in particular the visual word form area (VWFA) (Brem et al., 2020; Monzalvo et al., 2012; Shaywitz et al., 2002; Maurer et al., 2007; Boros et al., 2016; Kronbichler & Kronbichler, 2018). While dyslexia has been linked to both genetic and neurocognitive factors such as phonological abilities, its exact causal mechanisms remain unclear (Peterson & Pennington, 2015; Werth, 2023).

Advances in Machine Learning have significantly enhanced our ability to simulate and understand neural processes by leveraging biologically inspired computational frameworks. These developments have enabled the modeling of brain activity with increasing fidelity, capturing not only the structural characteristics but also the precise neural patterns of the human brain. Recent work has demonstrated that artificial neural networks can approximate the brain's mechanisms, aligning closely with neural responses observed in human cortical regions, such as in vision and language (Yamins & DiCarlo, 2016; Schrimpf et al., 2020; 2021).

---

[*]Joint supervision
[1]Code available via GitHub.

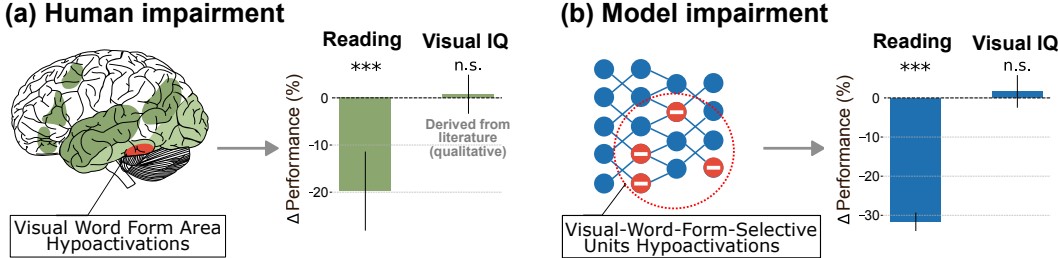

Figure 1: **Modeling dyslexia via visual-word-form hypoactivation. (a)** In humans, reduced activity in the visual word form area is thought to result in diminished performance on reading-related measures while sparing general visual intelligence. **(b)** Testing this hypothesis in vision-language models, we find that ablating visual-word-form-selective units produces the same dissociation.

Building on this foundation, our research aims to extend computational approaches of the healthy brain to the modeling of specific brain disorders — here focusing on dyslexia. In particular, we develop a computational framework to model dyslexia by functionally identifying and perturbing specific units within vision-language models. This approach simulates hypoactivations documented in the VWFA of dyslexic subjects (Brem et al., 2020), abstracting away genetic and other contributing factors. A successful model simulation of dyslexia should exhibit *selective* impairment of reading performance while preserving other cognitive functions such as general intelligence and reasoning.

We show that identifying and ablating visual-word-form-selective units in a vision language model (such as Qwen (Yang et al., 2024)) indeed leads to diminished performance on dyslexia screening assessments like the Rapid Online Assessment of Reading (ROAR) (Yeatman et al., 2021), while leaving general visual intelligence measures unaffected, including Raven's Progressive Matrices (Burke, 1958; Zhang et al., 2019) and the Kempler Test (Kempler et al., 1998) (Fig. 1). Ablating random units on the other hand does not have a selective effect on reading abilities and instead equally affects visual reasoning performance. An investigation into chosen hyperparameters reveals that the choice of layer type is critical while the effect varies more smoothly with the number and severity of affected units. Testing our model's reading deficits in more detail we find that it mirrors phonological deficits in human dyslexic subjects, but shows no significant impairment on orthographic stimuli. Overall, our computational results mimic empirical findings in dyslexic individuals, exhibiting selective reading-specific impairments without corresponding deficits in general intelligence (Snowling et al., 2020; Peterson & Pennington, 2015), and thus establish a computational framework for modeling the neural mechanisms underlying brain disorders.

## 2 BACKGROUND & RELATED WORK

**Behavioral effects of dyslexia.** Dyslexia is characterized by persistent difficulties in accurate and/or fluent word recognition and spelling, despite adequate education, normal intelligence, and intact sensory abilities (Lyon et al., 2003). These difficulties are closely linked to deficits in phonological awareness and processing, often manifesting as problems with decoding unfamiliar or nonsense words, rapid naming, and word finding. Further challenges may include variable difficulties with letter–sound learning, oral reading accuracy, written composition, and reading comprehension (Roitsch & Watson, 2019). Importantly, studies indicate that the core mechanisms of dyslexia are consistent regardless of IQ (Stanovich, 2005; Tanaka et al., 2011).

**Hypothesized neural substrate.** A growing body of evidence highlights the central role of the Visual Word Form Area (VWFA) in the neurobiology of dyslexia. The VWFA, located in the left ventral occipitotemporal cortex, is critical for fluent reading as it specializes in the rapid and automatic recognition of written words (Dehaene & Cohen, 2011; McCandliss et al., 2003). In individuals with dyslexia, this region often shows functional and anatomical abnormalities. Longitudinal studies have revealed that dyslexic readers consistently exhibit smaller and less selective VWFAs compared to typical readers, even after significant improvements in reading performance through targeted training (Mitchell et al., 2025; Brem et al., 2020). These persistent differences suggest that

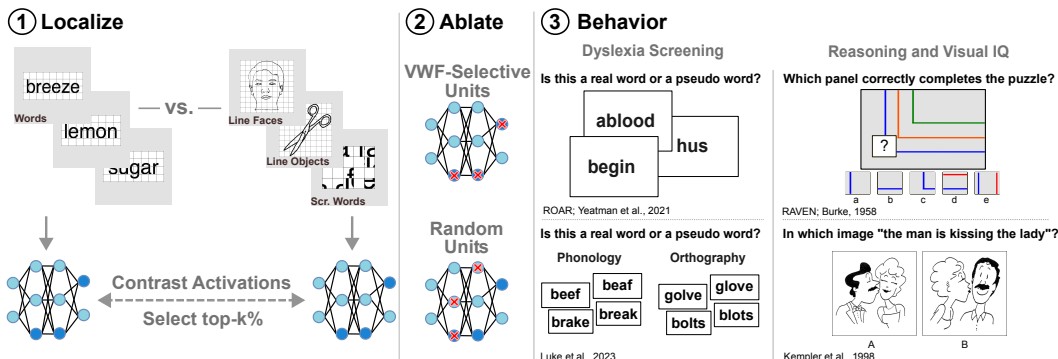

Figure 2: **Identifying visual-word-form-selective units in VLMs. (1)** To identify VWF-selective units, we compare unit activations in response to images of words versus images of non-words, and identify the units that exhibit the strongest word selectivity. **(2)** To model the reduced VWFA activity observed in dyslexic individuals, we ablate the localized units. As a control, we ablate an equal number of randomly selected units. **(3)** To assess the impact of ablations, we evaluate model performance on dyslexia screening tasks (ROAR (Yeatman et al., 2021) and the Lexical Decision benchmark (Luke et al., 2023)) as well as on visual IQ and reasoning tasks (RAVEN (Burke, 1958) and Kempler et al. (1998) sentence comprehension tasks).

VWFA atypicalities are not merely a consequence of poor reading experience but represent a stable neurobiological trait of dyslexia. Empirical studies consistently report VWFA hypoactivation and structural differences in dyslexics across languages and cultures (Paulesu et al., 2001; Silani et al., 2005), reinforcing its cross-linguistic relevance. Moreover, lesions in this region have been causally linked to acquired reading disorders (Turkeltaub et al., 2013), underscoring the VWFA's essential role in both typical and atypical reading processes. Nevertheless, the causal status of VWFA abnormalities remains debated, with some studies suggesting that these differences may reflect a consequence rather than a primary cause of dyslexia (Olulade et al., 2013; Valdois, 2010).

**Models of Brain Function.** Computational neural networks have proven effective at predicting both behavioral performance and neural responses in healthy subjects, making them valuable tools for studying brain function. In the visual domain, deep convolutional as well as transformer-based models have successfully captured neural activity in the primate ventral visual stream across object and scene recognition tasks (e.g., Yamins et al., 2014; Khaligh-Razavi & Kriegeskorte, 2014; Schrimpf et al., 2018; 2020; Cadena et al., 2019; Spoerer et al., 2020; Zhuang et al., 2021; Wang et al., 2023; Margalit et al., 2024; Gokce & Schrimpf, 2024; Lonnqvist et al., 2025; Tang et al., 2025). In the language domain, transformer-based and recurrent models have accurately predicted neural responses related to semantic, syntactic, and phonological processing (e.g., Schrimpf et al., 2021; Caucheteux et al., 2022; Goldstein et al., 2022; Toneva et al., 2018; Hosseini et al., 2024; Aw et al., 2024; Tuckute et al., 2024; Rathi et al., 2025; AlKhamissi et al., 2025; Du et al., 2025).

Machine learning methods thus provide a powerful platform for capturing the cognitive and neural patterns associated with dyslexia. By enabling controlled experiments and hypothesis testing in silico, such models can offer new insights into the mechanisms underlying the disorder and might support the development of targeted diagnostic and intervention strategies.

Another approach to model brain disorders employs connectivity-based models, which simulate large-scale brain dynamics using empirically derived structural or functional connectomes. In the context of reading and dyslexia, such models have been used to relate altered white matter pathways and disrupted functional connectivity to deficits in phonological processing and reading performance (Müller et al., 2017; Sihvonen et al., 2021), but these comparatively coarse-grain approaches remain limited in capturing the precise underlying neural mechanisms.

To the best of our knowledge, there is no prior work on modeling brain disorders using system-level neural models, particularly in the context of dyslexia. While there are studies that employ computational models to investigate aspects of dyslexia, such as visual information processing (Ogawa et al., 2023) and handwriting anomalies (Alevizos et al., 2024), these approaches do not simulate the complex neural activity changes associated with dyslexia. Therefore, our work represents a first

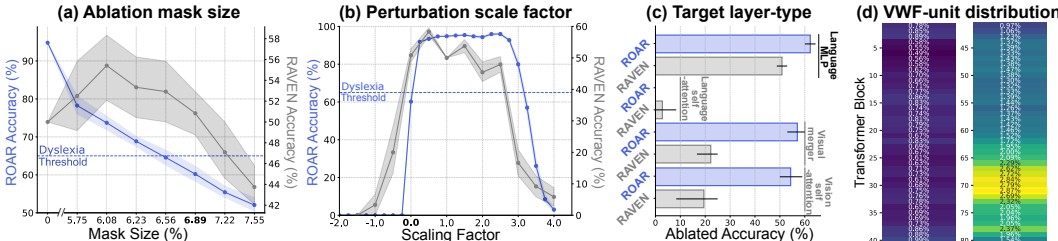

Figure 3: **Ablating visual-word-form-selective units in the model. (a)** Increasing the number of VWF ablated units translates into a severe monotonic performance decline in ROAR (blue), while RAVEN (gray) is only affected at larger mask sizes. We chose the first mask size (bold) where ROAR performance falls below the dyslexia threshold (blue dashed line). Shaded regions represent 95% confidence intervals. **(b)** Beyond full ablation (bold), scaling unit activity (with a fixed mask size of 6.89%) has little effect for positive scaling, while negative scaling severely degrades outputs non-selectively. **(c)** While ablations in all layer types substantially affect performance, only the MLP components of the language decoder showed selective effects, indicating its core involvement in reading (full trends in Fig. 11). **(d)** Distribution of VWF-selective units across the 80 transformer blocks of the language decoder. Ratios are mean across 20 random seeds and resampling of the localizer stimuli; standard deviations never exceed 0.03%.

attempt to model brain disorders (here, dyslexia) through hypothesized neural activity changes in state-of-the-art models.

## 3 BENCHMARKS

We evaluate model performance using a suite of standardized assessments originally designed for human subjects in clinical psychology and cognitive neuroscience (Fig. 2 right). These include tests of lexical decision-making (ROAR, Yeatman et al., 2021), visual reasoning (RAVEN, Burke, 1958; Zhang et al., 2019), sentence-level syntactic comprehension (Kempler Test, Kempler et al., 1998), and dissociations between orthographic and phonological processing (Luke et al., 2023). By repurposing these human-designed benchmarks for VLMs, we enable a targeted and comparable evaluation of specific cognitive capacities and their impairments, providing a structured framework for interpreting model behavior in relation to established cognitive theories.

**Rapid Online Assessment of Reading (ROAR)** is a browser-based, self-administered lexical decision task that evaluates core reading skills by measuring both accuracy and speed in distinguishing real words (e.g., "able", "animal") from pseudo words (e.g., "ablood", "accastanct") (Yeatman et al., 2021; more examples in Table 1). In our model, however, response time is not a consideration, so we evaluate performance solely based on the accuracy of lexical decisions. On each trial, the model is shown an image of a letter string and must decide whether it is a real word or a pseudo word. We present 200 real words and 200 pseudo words drawn from the full ROAR-Word corpus as the train set for finding the minimal VWFA mask, with the remaining 50 real words and 50 pseudo words serving as the test set for lexical evaluation. *Dyslexia Threshold*: We define 65% ROAR performance as the threshold at which subjects are considered as dyslexic. This number is one standard deviation below the mean ROAR-score of the human population, consistent with prior works in humans (Wagner et al., 2020). This choice is also meaningful from an epidemiological perspective as dyslexia is estimated to affect about 5–20% of people (Wagner et al., 2020), and this prevalence aligns well with scores in the range of $[1.65, 0.95]$ standard deviations below the mean.

**Raven's Progressive Matrices (RAVEN)** is a nonverbal assessment of fluid intelligence in which participants must select, from five candidate panels, the one that correctly completes a 3×3 or 2×2 matrix by preserving an underlying pattern of shapes or spatial relations (Burke, 1958; Zhang et al., 2019; examples in Appendix Fig. 9). Because it minimizes linguistic and cultural biases, RAVEN serves as an ideal control for general visual-spatial reasoning in our study. In our experiments, we administered the easy section of the RAVEN clinical version which consists of twelve items. By assessing the model's accuracy on these twelve puzzles, we establish a baseline for general visual intelligence against which any reading-specific impairments can be contrasted.

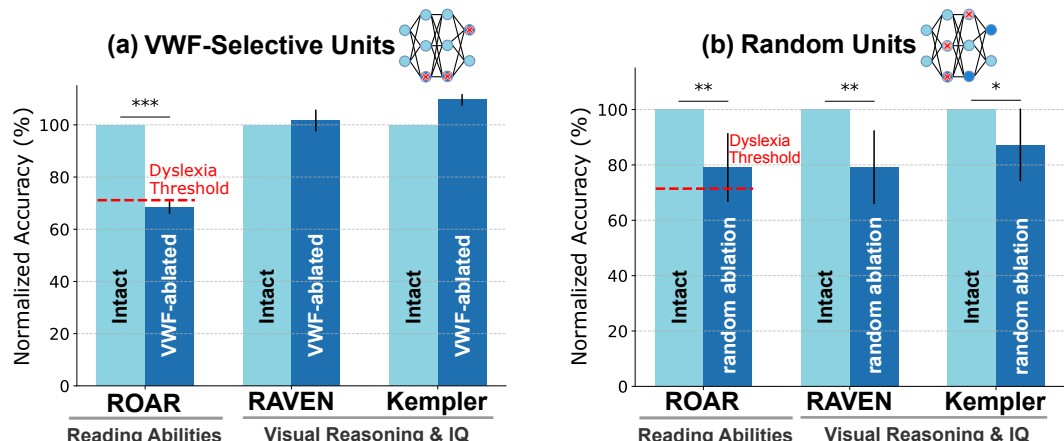

Figure 4: **Reading-selective deficits from ablating VWF-selective units.** **(a)** Ablating VWF-selective units led to a selective reading deficit below the dyslexia threshold (ROAR, $p < 0.012$), while performance on visual IQ and reasoning benchmarks (RAVEN, Kempler) remained intact or was slightly enhanced. **(b)** Ablating an equal number of randomly selected units from the same layers affected performance throughout, with ROAR remaining above the dyslexia threshold and significant impairments to visual reasoning. Dark blue bars indicate ablated model accuracy, relative to the intact model (light blue bars). Error bars denote 95% confidence intervals, and significance was assessed with one-sample, one-tailed Student's t-test.

**Kempler's Sentence Comprehension Test** is a clinical assessment designed to evaluate syntactic comprehension by presenting participants with pairs of images accompanied by a spoken or written sentence (Kempler et al., 1998). The task is to choose the image that correctly corresponds to the sentence, making it a useful probe of grammatical understanding and sentence-level semantics. In our implementation, we adapt this test for model-based evaluation by using only the image pairs from the original stimulus set, while providing the sentence as a text prompt. The model is asked to decide which image matches a given caption (examples in Appendix Fig. 10). This transforms the task into a visual question answering (VQA) challenge, allowing us to assess the model's capacity for sentence comprehension grounded in visual reasoning.

**Lexical Decision with Orthographic and Phonological Manipulations.** To further dissect reading impairments into orthographic and phonological components, we incorporated stimuli from a lexical decision task developed by Luke et al. (2023). This benchmark includes four stimulus types: (1) homophones (e.g., "brake" and "break"), (2) pseudo-homophones (e.g., "beaf" and "birf"), (3) transposed-letter (TL) neighbour words (e.g., "blots" and "bolts"), and (4) TL non-words (e.g., "golve" and "glove"). These stimuli were originally used to contrast phonological and orthographic theories of dyslexia in human readers. For analysis, we group the stimuli into two categories: phonology-sensitive (homophones and pseudo-homophones) and orthography-sensitive (TL words and TL non-words). The model performs a lexical decision on each item (an image containing a word or a non-word), and accuracy serves as the primary outcome measure. By comparing performance across these two groups, we can determine whether the model finds phonologically demanding items or orthographically demanding items more challenging.

## 4 METHODOLOGY

**Functional Localization of VWF-selective Units.** Given the well-established role of the VWFA in dyslexia, we target visual-word-form-selective (VWF-selective) units in vision-language models (VLMs) as regions of interest for artificial lesions. To reflect the dual involvement of visual and linguistic processes in dyslexia, we focus on computational models that process images (and text) as input which reflect the full processing chain from retina (pixel) input to language. These VLMs, by presenting visual stimuli as image input and task instructions as text tokens, allow us to dissociate visual processing impairments from general language deficiencies, and their integration of visual and linguistic modalities within a single architecture provides a unified framework for examining the

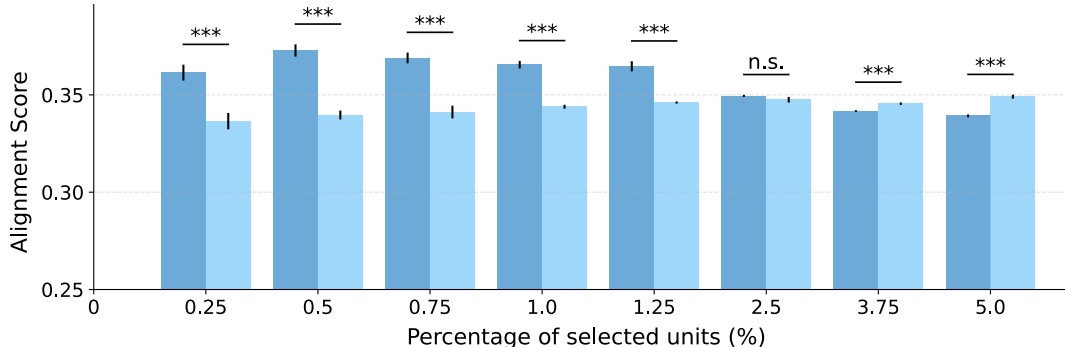

Figure 5: **VWF-selective units align with human VWFA activity.** Alignment between predicted and actual fMRI responses in the human VWFA (noise-normalized), computed across the top-k% of VWF-selective units of the model (dark blue bars) and randomly selected units (light blue bars) for 5 seeds. At small subset sizes, VWF-selective units show significantly higher alignment than the average of five randomly selected subsets. Error bars indicate 95% confidence intervals across seeds.

cortical processes underlying word recognition and reading impairments in dyslexia. Our method enables a hypothesis-driven investigation of reading-specific mechanisms directly within artificial systems and provides a scalable and manipulable alternative to costly human studies.

Using functional localizers to identify selective units in computational models is a neuroscience-inspired approach rooted in studies like Kanwisher et al. (1997), who used fMRI localizers to identify specialized brain regions. Saygin et al. (2016) functionally localized the VWFA by contrasting responses to written words with visually matched control stimuli, including line drawings of faces, scrambled words, and line drawings of objects, enabling robust identification of word-selective regions in the ventrotemporal cortex. Recently, AlKhamissi et al. (2025) adapted such functional localization methods for large language models and identified the causal role of language-selective units through targeted ablations. We adopt a similar approach by applying a functional localizer paradigm to our vision language model, targeting VWF-selective units.

Specifically, we use a classic fMRI localizer from neuroscience (Saygin et al., 2016) to identify VWF-selective units in computational models. The model is presented with the four stimulus categories from Saygin et al. (2016): written words, scrambled words, faces, and objects. For each considered model unit, we compute a t-statistic comparing responses to word images versus the three non-word control categories. The t-statistic quantifies how strongly a unit prefers words relative to other stimuli by measuring the difference in response means normalized by response variability. Units with higher t-statistics respond more selectively and reliably to words, helping us identify candidate VWF-selective units in the model. Units are then ordered by descending t-statistic, and we define the top $k\%$ of these units to be the model's VWF-selective units (Fig. 2).

**Minimal Subnetwork for Dyslexia Simulation.** To induce a dyslexia-like impairment, we gradually increase the fraction of the top k% VWF–selective units that are ablated (i.e., set to zero) within the model's language layers. We begin with no masking (0%) and repeatedly mask a larger proportion of these units, evaluating at each level the model's accuracy on the train subset of ROAR stimuli. We stop increasing the mask size as soon as the ROAR score falls below 65%, which we defined as the ROAR dyslexia threshold (Section 3). The smallest proportion of masked units that meets this criterion defines our VWF–selective units (see also Fig. 3a).

**Model Details.** We focused our analyses on `Qwen2-VL-72B` (Yang et al., 2024), chosen for its strong OCR and visual understanding performance. Qwen2-VL-72B is a large-scale autoregressive transformer integrating visual and textual modalities. Within the model, we examined the MLP layers of the language decoder where the minimal subnetwork for dyslexia corresponds to approximately 6.89% of all units (see below for hyperparameter analyses). We successfully elicited reading deficits in all tested models (additionally `Molmo-72B`, Deitke et al., 2024; and `PixTral-12B`, Agrawal et al., 2024). Deficits were reading-specific, with no significant effects on visual IQ benchmarks (Appendix Fig. 8).

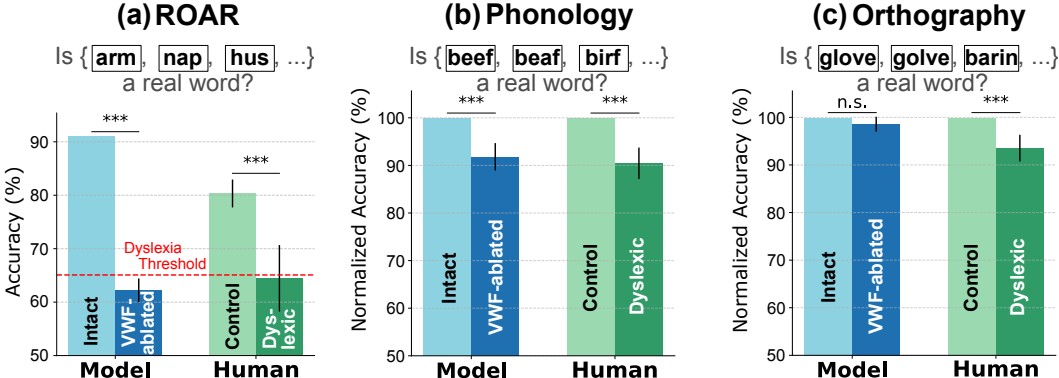

Figure 6: **Model mirrors human phonological reading deficits.** **(a)** Lexical decision accuracy on the ROAR reading test before (light blue) and after (dark blue) ablation of VWF-selective units. Following ablation, model lexical decision accuracy at real/pseudo word classification drops below the dyslexia threshold, paralleling dyslexic participants (dark green) relative to control subjects (light green). **(b)** Ablated model performance decreases for phonologically confusable stimuli (e.g. "beaf" which sounds the same as "beef" but is a pseudo word), indicating a phonological deficit and mirroring observations in humans. **(c)** Ablated model performance is not significantly affected on orthographically confusable stimuli (e.g. "golve" which looks similar to "glove" but is a pseudo word), whereas dyslexic humans tend to be affected.   Error bars denote 95% confidence intervals; model results are averaged over 20 random seeds, each corresponding to a different sample of the localizer; significance assessed via one-tailed Student's t-test (Appendix A.4).

**Perturbation Strength.** Beyond a full ablation (activity = 0), we additionally experimented with parametric perturbations of VWFA units by varying the scaling factor on the activations of the selected VWF-selective units. Specifically, we tested scaling factors in the range $[-2, 4]$, including sublinear, neutral (1.0), and superlinear amplification. We observed that only complete ablation (scaling factor = 0) reliably produced the selective impairments characteristic of dyslexia (Fig. 3b). Perturbations at other scales either had negligible behavioral effects or resulted in severely degraded output coherence, producing nonsensical or empty responses (Table 3). Consequently, we focus our analysis on full ablation as the most faithful computational analogue of focal VWFA disruption.

**Layer Choice.** To identify VWF-selective units, we evaluated subnetworks from different layer types across the model. The MLP gate projection layers in the language decoder emerged as the most selective, aligning with prior findings that MLP layers exhibit strong knowledge-specific selectivity (Meng et al., 2022; Zhang et al., 2022). To support this interpretation, we ablated minimal subnetworks from other components of the model, the vision encoder (`visual.blocks.{i}.attn.proj`), the visual merger (`visual.merger.mlp.{i}`), and the language decoder's self-attention outputs (`model.layers.{i}.self_attn.o_proj`). These ablations led to sharper declines in RAVEN performance relative to ROAR performance, indicating that the observed effects are not purely reading-selective (Fig. 3c). Based on these results, we define VWF-selective units as those located in the MLP layers (`model.layers.{i}.mlp.gate_proj`) across all 80 transformer blocks of the language decoder (Fig. 3d).

## 5 RESULTS

We evaluated the impact of VWFA-targeted ablation on model performance across tasks that probe reading, language comprehension, and nonverbal reasoning. We examine how selective disruption of functionally localized units affects different cognitive domains.

**Selective Reading Deficits, Preserved Reasoning Capabilities.** Ablating visual-word-form (VWF)-selective units leads to a substantial decline in lexical decision accuracy on the ROAR lexical reading task (Fig. 4a). Table 1 shows example responses of the ablated model. Performance in fact dropped below the dyslexia threshold on the held-out test set, a marked reduction from the model's

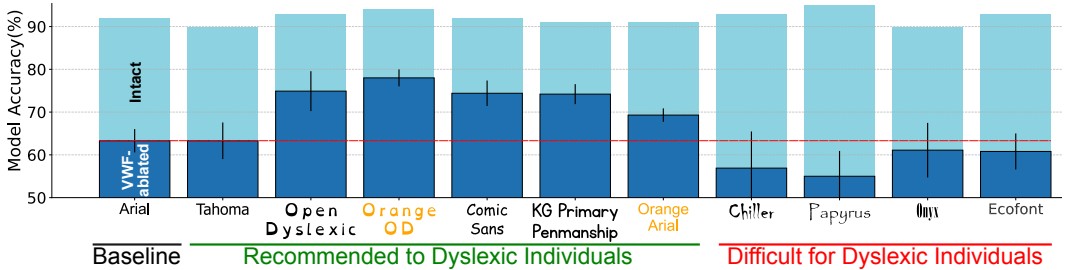

Figure 7: **Model predicts font-specific reading performance.** Accuracy on ROAR lexical decision items across fonts for intact (light blue) and ablated "dyslexic" (dark blue) model, averaged over 10 random seeds. The dyslexic model shows significantly improved performance for OpenDyslexic, OpenDyslexic on an orange background, Comic Sans, and KG Primary Penmanship ($p \ll 0.001$), as well as for Orange Arial ($p \ll 0.01$), while performance is significantly worse for Papyrus ($p \ll 0.01$). In contrast, the intact model's accuracy remains stable across fonts with no significant differences. Error bars denote 95% confidence intervals across seeds.

pre-ablation accuracy ($-32\%$ in normalized accuracy with $p \ll 0.01$ using a one-tailed Student's t-test for whether ablated model performance was significantly lower than baseline performance on each of the tasks; see Appendix A.4 for more details). In contrast, performance on the RAVEN's Progressive Matrices task remained stable ($p \approx 0.75$), indicating that the lesioned subnetwork selectively impairs reading ability while sparing broader visual reasoning processes. We further evaluated sentence comprehension and visual grounding using the Kempler Sentence Comprehension Test as another visual Q&A task. Following ablation, the model's accuracy on the Kempler task did not decrease compared to the non-ablated baseline ($p \approx 1$, one-tailed test; a two-tailed Student's t-test reveals a statistically significant increase in performance of $+10\%$, $p \ll 0.01$). This pattern parallels empirical findings in some dyslexic individuals, where deficits in phonological or orthographic processing are accompanied by preserved or even enhanced performance in visual-spatial reasoning and comprehension tasks (von Károlyi et al., 2003; Stanovich, 2005; Tanaka et al., 2011).

**Ablating Random Units Does Not Lead to Selective Reading Impairments.** To verify that the observed dyslexia-like effects were not simply due to the number or layer-wise distribution of ablated units, but rather depended on the specific identity of the VWF-selective units, we conducted a control experiment. We preserved both the total number of ablated units and the selected layers, but randomly selected which specific units were ablated within each layer. This approach did not lead to selective reading impairments: the performance on *all* benchmarks dropped ($-21\%$ with $p < 0.003$ for ROAR, $-21\%$ with $p < 0.004$ for RAVEN, and $-13\%$ with $p < 0.042$ for Kempler), and reading performance did not drop below the dyslexia threshold ($p \approx 0.87$; Fig. 4b). Furthermore, when the same number of units was ablated but distributed randomly across the entire network rather than within the same layers, the model's outputs largely degraded, often generating incoherent or empty responses, confirming that the reading-specific impairment depends on targeting VWF-selective units.

**Human-Like Phonological Deficits.** VWFA-targeted ablation selectively impairs reading performance, producing deficits reminiscent of those observed in dyslexic individuals. To assess whether the observed deficit was primarily phonological or orthographic, we evaluated model performance on the lexical decision benchmark from Luke et al. (2023). Post-ablation, model accuracy dropped significantly on phonology-sensitive items (Fig. 6; $-8\%$ with $p \ll 0.01$), but remained relatively stable for orthography-sensitive stimuli ($p > 0.059$) indicating disproportionate disruption of phonological processing with preserved orthographic representations. Human behavioral data released by Luke et al. (2023) reveal a related dissociation: Dyslexic participants showed a pronounced accuracy reduction on phonology-sensitive items ($-9\%$ with $p \ll 0.01$), but also demonstrated impairments on orthography-sensitive items($-6\%$ with $p \ll 0.01$) *on average*. Dyslexia often co-occurs with other learning and language disorders (Nicolson & Fawcett, 2011; Chalikia et al., 2025), yet the available dataset does not report such metadata, which may contribute to heterogeneity and the observed orthographic impairments. The dominant view holds that dyslexia arising from VWFA-related impairments is primarily phonological, consistent with the selective phonological deficit observed in our ablated model.(McCandliss et al., 2003).

Table 1: **Common error patterns of the VWF-ablated model.** *Blank*: Model gave no output at all (completely empty response). *Misclassification*: Model explicitly labeled a real word as pseudo, or a pseudo word as real. *Contextual Over-Interpretation*: Model invoked external references (languages, acronyms, domains) to justify the wrong answer. *Ambiguous Hedging*: Model refused to commit, saying the word could be real or pseudo depending on context. *Gibberish / Corrupted*: Output was meaningless fragments, random characters, non-English text, or truncated nonsense. More examples of each pattern in Table 2.

| Response type | Sample Word Image | Example Model Output |
|---|---|---|
| Blank | `accustomed` (real) | – |
| Misclassification | `yammerring` (real) | The word "yammerring" looks like a pseudo word. |
| Contextual Over Interpretation | `hus` (pseudo) | The word "hus" in the image is a real word in English. It is a noun that means "husk" which is a type of edible corn. |
| Ambiguous Hedging | `dood` (pseudo) | The word "dood" appears in the image. It seems to be a pseudo word because it has no meaning in English. However, it could potentially be a real word if it were part of a specific context or phonetic representation, for which further investigation would be necessary. |
| Gibberish / Corrupted | `imeyits` (pseudo) | image of the image of the image of the image of the following. |

**Alignment Between VWF-selective Units and Human Brain Activity.** Building on the observed behavioral alignment, we next assessed whether selected units correspond to human neural activity. We quantified how VWF-selective units align with human neural responses using a standard encoding-model framework (Naselaris et al., 2011; Schrimpf et al., 2021). For each subject in the Marvi et al. (2025) dataset, ridge-regression models were trained to predict voxel-wise fMRI responses from the model's video-evoked activations using 5-fold cross-validation. Pearson correlations between predicted and actual voxel responses were averaged across folds and divided by each voxel's noise ceiling to yield the alignment score. We evaluated model alignment using subsets of the most VWF-selective units, varying the subset size from 0.25% to 5% of all units. For each subset size, we generated five VWF-selective masks by resampling the localizer stimuli with five different seeds. Each VWF-selective mask was paired with a size-matched random mask for comparison. At smaller percentages, VWF-selective subsets showed significantly higher neural alignment than random subsets, indicating that these units encode brain-relevant structure rather than arbitrary features and supporting their role in both behavioral and neural correspondence to visual word-form processing (See Fig. 5). For larger percentages (above 1.25%), the same effect is not observed. This likely reflects differences in feature dependence; VWF-selective subsets are drawn from a ranked list and therefore contain correlated units, whereas random subsets contain more heterogeneous, less correlated features. As more units are included, these independent random features improve fMRI prediction, while correlated VWF-selective features add redundant information. This suggests that VWF-selective units form a specialized subspace that is highly informative when the most selective units are sampled.

**Predicting Dyslexia-Friendly Fonts.** Because the ablated model reliably reproduces behavioral patterns observed in dyslexic readers, it can be used to evaluate reading interventions and design supportive visual stimuli. To illustrate this, we presented both the intact and ablated models with the original ROAR lexical decision items in Arial (the standard font used in the benchmark), keeping the font size as similar as possible, and then systematically replaced the font with alternatives known from prior work and expert recommendations to be either challenging or easier for dyslexic

readers. Fonts tested included, Tahoma, OpenDyslexic (Gonzalez, 2014), Comic Sans, KG Primary Penmanship, Chiller, Papyrus, Onyx, and Ecofont. We also tested Arial and OpenDyslexic on a dark orange background, grounded in personal reports from dyslexic individuals that colored backgrounds can aid reading. The ablated model mirrored human sensitivity and performance decreased or remained unchanged on fonts considered difficult, and improved or remained stable on dyslexia-friendly fonts, whereas the intact model showed no systematic differences (See Fig. 7).The ablated model showed significant improvement in lexical decision accuracy relative to Arial for OpenDyslexic, OpenDyslexic on an orange background, Comic Sans, and KG Primary Penmanship ($p \ll 0.001$), and for Orange Arial ($p \ll 0.01$). In contrast, performance significantly decreased for Papyrus ($p \ll 0.01$) compared to Arial. This demonstrates that the model can capture font-specific reading difficulties observed in dyslexic readers. Importantly, this approach opens a novel possibility unique to computational modeling. Future work can design or optimize new fonts by identifying visual configurations that maximize performance in the ablated model while preserving readability in the intact model.

## 6 DISCUSSION

By inducing dyslexia-like behavior in vision language models, we can examine how specific changes affect performance and study the mechanisms involved. These models, though abstracted from biology, preserve key correspondences with brain-like processing – enabling insight into brain disorders, prototyping interventions, and overcoming limitations of human data collection.

**Interpretability.** This work advances interpretability by identifying a reading-selective subnetwork in VLMs reminiscent of the VWFA in humans. Targeted ablations within this subnetwork induce a phonological rather than an orthographic deficit, despite the model never processing auditory input. These findings suggest that functional localization can be used to uncover mechanistic and causal links between specialized neural circuits and behavioral outcomes. We also observe a performance increase on nonverbal benchmarks post-ablation, paralleling findings in some human dyslexics (von Károlyi et al., 2003; Lam & Tong, 2021), which we plan to explore further in future work with the goal of enhancing learning via ablation of non-selective units.

**Deficit Characterization.** Hypoactivation of the VWFA is often linked to phonological deficits (McCandliss et al., 2003). By ablating VWFA-selective units, we find that the resulting deficit is phonological, supporting the dominant view that phonological dysfunction is primary, at least within the model.

**Broader Applicability.** Despite differences from biological brains, brain-aligned models provide useful proxies for identifying mechanisms underlying behavioral deficits. Models exhibiting dyslexia-like behavior can act as digital twins, enabling controlled causal experiments (e.g., targeted ablations) that are difficult in humans. The proposed localization–ablation–behavioral testing framework generalizes to other brain disorders where contrast stimuli identify neural substrates and behavioral benchmarks assess causal effects, offering a flexible platform for studying psychiatric conditions beyond dyslexia and informing intervention strategies such as dyslexia-aware font design to improve reading performance in dyslexic readers.

## 7 CONCLUSION

By localizing and lesioning visual-word-form-selective units in vision-language models, this work provides a biologically inspired simulation of dyslexia, capturing its core reading-specific deficits while preserving broader cognitive functions. Our findings mirror empirical dissociations observed in dyslexic individuals and validate the functional relevance of the visual word form area. Our approach demonstrates how mechanistic manipulations of artificial neural network models can simulate selective cognitive deficits, offering a novel computational framework for testing hypotheses about the causal role of specific neural circuits in reading. Via the identification of model components whose ablation mimics dyslexic-like impairments, such in-silico experiments could help test hypotheses about neural targets for early screening, suggest biomarkers for subtyping dyslexia (e.g., based on differential phonological versus orthographic deficits), and inform the development of targeted intervention strategies such as dyslexia-friendly fonts. This framework lays the groundwork for modeling not just the neurotypical but also the neurodivergent brain.

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

## A  APPENDIX

### A.1  OTHER MODELS

We applied our VWFA localization and ablation methodology across two other vision-language models: Molmo-72B(Deitke et al., 2024), and PixTral-12B(Agrawal et al., 2024). Molmo-72B showed a baseline ROAR accuracy of $83\%$, providing a suitable foundation for evaluating selective impairments. After ablating the VWFA-analogous units, ROAR performance exhibited a statistically significant decrease, whereas RAVEN performance did not show a statistically significant change, indicating that the observed deficit is specific to reading-related processing (Fig. 8a). PixTral-12B demonstrated robust baseline performance on ROAR: $86.75\%$. After ablating the VWFA-selective units, ROAR performance decreased, whereas the change in RAVEN performance was not statistically significant, suggesting that the method can be generalized to other models.

### A.2  VISUAL IQ AND REASONING BENCHMARKS

Figure 9 shows an example of the non-verbal benchmarks. More examples of Kempler are shown in Figure 10.

### A.3  PROMPTS USED ACROSS ALL EXPERIMENTS

For transparency and reproducibility, we include the exact prompts used across all lexical, phono-logical, and nonverbal reasoning benchmarks. These prompts were created with the explicit goal of mimicking how human participants are typically instructed to perform these tasks, ensuring that the model receives directions that parallel standard experimental protocols.

**Activation Extraction Prompt.**

```
Describe the image.
```

**Lexical Decision Prompt.**

```
A real or pseudo word will be presented to you in
an image.  The pseudo words might look like English
words, but they don't mean anything in English.  For
example, laip, bove or cigbert are pseudo words.  The
real words will be ones you recognize.  They are real
English words like is, or basket, or lion.  Please
answer the following question:  Is the word in the
image a real word or a pseudo word?
```

**KEMPLER Prompt.**

```
In which image [DESCRIPTION OF TARGET IMAGE]? After
providing the reason, give your final answer in this
format:  "The answer is picture a" or "The answer is
picture b".
```

**RAVEN Prompt.**

```
You will be presented with a nonverbal Raven's
Progressive Matrices IQ puzzle.  The puzzle will
consist of a visual pattern with one missing element,
indicated by a question mark.  Your task is to
identify which of the five provided options best
completes the pattern.  Please choose the option that
logically fits the sequence.  Write the answer in one
digit in [1, 2, 3, 4, 5].
```

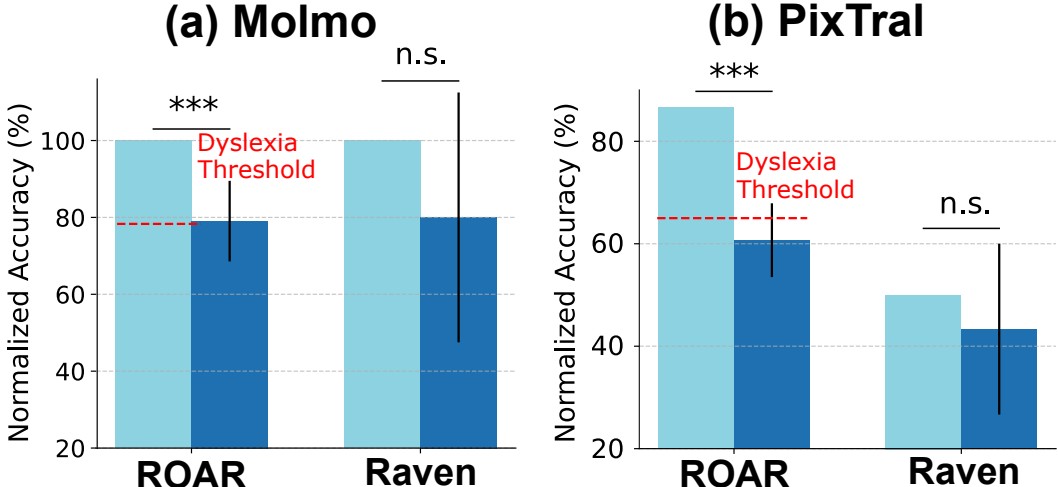

Figure 8: **Ablating VWFA-selective units in other vision-language models.** Plots show accuracy of the intact and ablated models on the ROAR and RAVEN benchmarks. Bars indicate mean accuracy over 5 random seeds, with error bars representing 95% confidence intervals. For Molmo-72B **(a)**, ablation of selective units significantly reduces ROAR performance from 83%, while RAVEN performance remains unchanged. For PixTral-12B **(b)**, ablation significantly reduces ROAR performance from 86% without affecting RAVEN, indicating that the deficits are selective for reading.

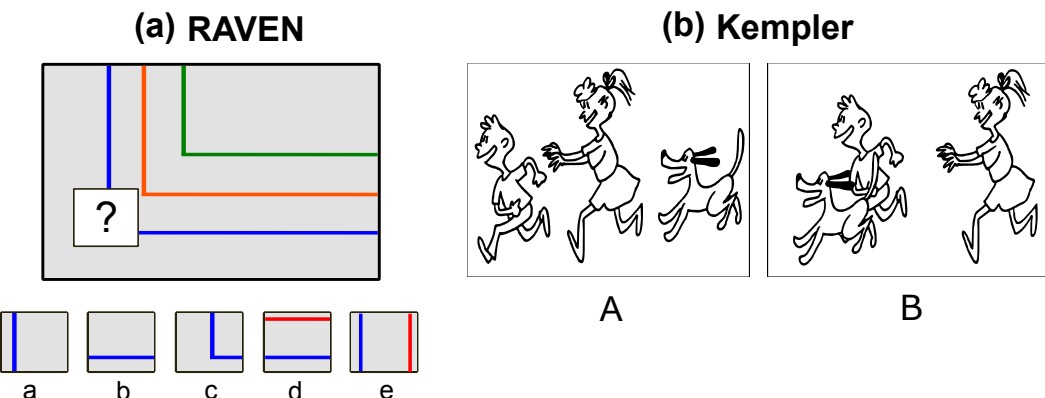

Figure 9: **Examples of IQ and visual reasoning benchmarks (a)** Example item from the RAVEN visual reasoning test. The task is to identify the panel (from five candidates) that correctly completes the matrix by following the underlying visual pattern. In this example, the correct answer is Option 3. RAVEN serves as a measure of general visual-spatial reasoning, independent of language ability. **(b)** Example item from the Kempler's Sentence Comprehension Test used in our model evaluation. Given a caption and a pair of images (A: left, B: right), the model is prompted to identify which image correctly depicts the described event. In this example, the model is asked: "In which image (A or B) the dog that chases the girl is following the boy?" (correct answer option A.)

## A.4 STATISTICAL TESTING

All model results are reported as averages over 20 random seeds, each corresponding to a different random sample of the localizer stimuli. To evaluate whether ablations reduced performance, we compared the 20 ablated values (one per seed) against the intact model's baseline using a one-tailed Student's $t$-test. The null hypothesis was that mean ablated performance equals baseline, and the alternative was that mean ablated performance is lower. A one-tailed test was chosen because our hypothesis was directional: ablations were expected to impair, but not improve, task performance. Reported $t$-statistics and $p$-values reflect this comparison for each benchmark. Intriguingly, a two-tailed Student's $t$-test, testing bidirectional changes of value distributions, revealed a statistically significant increase on the Kempler visual reasoning task.

## A.5 COMPUTATIONAL RESOURCES

All experiments were run on two NVIDIA A100 GPUs with 80GB memory each. The ablation implementation does not introduce meaningful overhead.

## A.6 USE OF LLM ASSISTANCE

We used *ChatGPT (GPT-5, OpenAI, 2025)* to aid in structuring and polishing the writing of this document. The authors reviewed, edited, and are fully responsible for the final content.

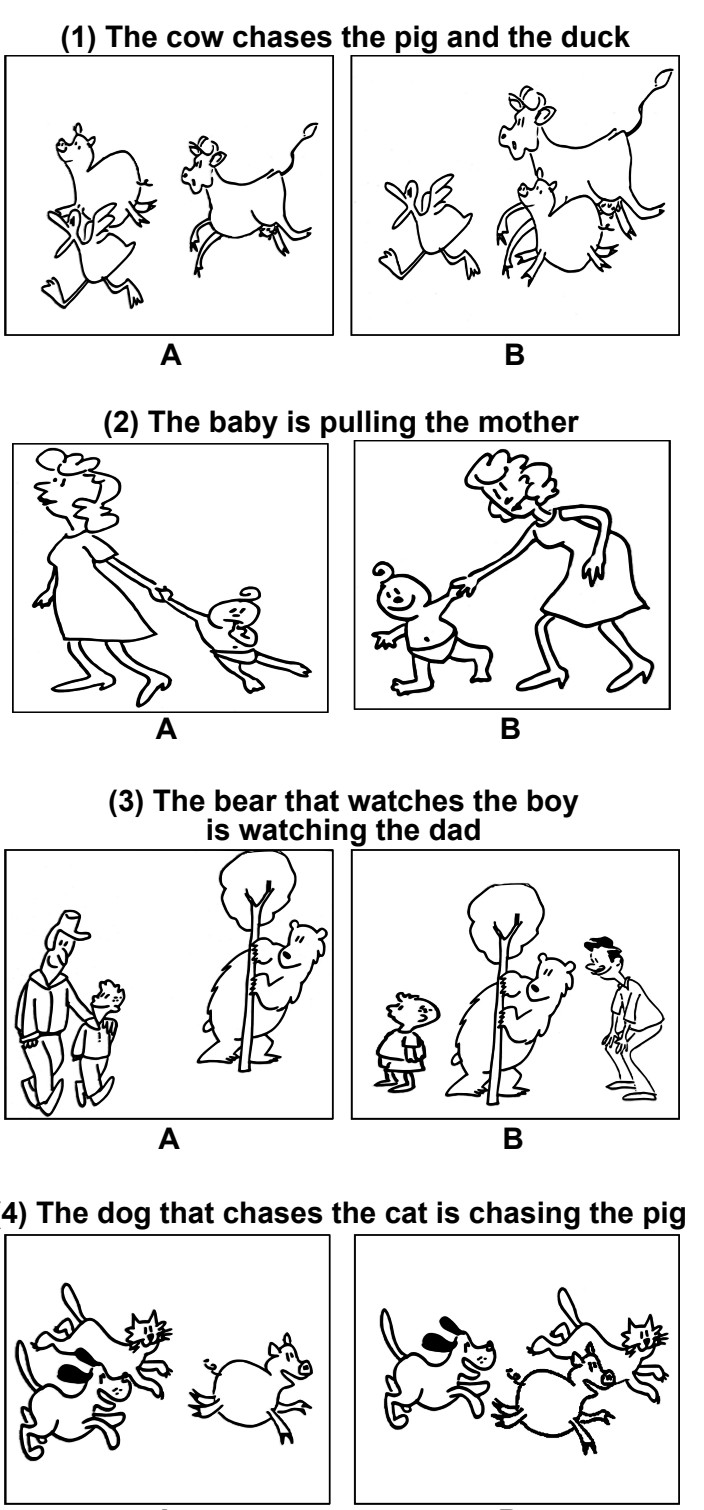

Figure 10: **Examples of IQ and visual reasoning benchmarks** Several items from Kempler's Sentence Comprehension Test, which evaluates the ability to map linguistic descriptions onto visual scenes. In each case, the model is given a caption and two candidate images (labeled A: left, B: right) and must decide which picture matches the sentence. Shown here are four sample items with increasing syntactic complexity. The correct answers for Items 1–4 are: A, B, A, B.

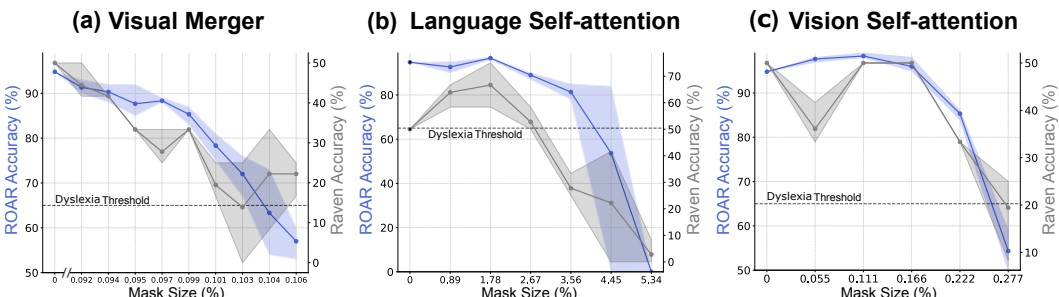

Figure 11: **Ablating units in non-selective layers does not result in selective impairments.** Testing other layers in the **(a)** visual merger (projection layers), **(b)** language decoder, and **(c)** vision encoder shows that RAVEN performance declines faster than ROAR performance indicating the effect is not reading-selective. Intriguingly, the self-attention layers show an inverse effect, with potential selectivity for RAVEN over ROAR. Shaded regions around each curve represent the 95% confidence intervals.

Table 2: **Further examples of wrong model responses to ROAR when VWF-selective units are ablated.** (Continued from Table 1.) Errors span misclassification, contextual over-interpretation, ambiguous hedging, and gibberish/corrupted outputs.

| Response type | Sample word image | Example model output |
|---|---|---|
| Misclassification | charring(real) | The word in the image is a pseudo word. |
| | creat(pseudo) | The word "creat" is a real word in English. It can be used as a verb, such as "to creat something." |
| | rackle(pseudo) | The word in the image is a real word. |
| Contextual Over-Interpretation | che(pseudo) | The word "che" is a real word in English. It is used as a conjunction in Italian grammar and also as a noun in English, referring to a type of cheese. |
| | hu(pseudo) | The word "hu" can be considered a real word in some contexts. It is used in Hungarian names and titles, such as Budapest University of Economics (HU). |
| | mich(pseudo) | The word "mich" in the image appears to be a real word... It seems to be a play on the word "Michigan." |
| | albust(pseudo) | The word "albust" is a real word in English, but it doesn't have any meaning associated with it. It appears to be a play on the word "albus." |
| Ambiguous Hedging | bi(pseudo) | The word "bi" can be a real word or a pseudo word depending on the context. It could mean "two" as a prefix, but alone it may not count as a word. |
| | ase(pseudo) | The word "ase" can be interpreted as a real word or a pseudo word depending on its context. If it stands alone, it is a pseudo word, but in words like "phrase" or "database" it is real. |
| | yos(pseudo) | The word "yos" appears to be a real word. It seems to be a part of a sentence or phrase, suggesting it could be meaningful in English. However, its full meaning is unclear. |
| Gibberish / Corrupted | napery(real) | (Chinese characters). |
| | pumer(pseudo) | (Chinese characters) |
| | cowbim(pseudo) | real words will bepseudo words will bepseudo words |
| | seashell(real) | the image of the image words |

Table 3: **Examples of wrong model responses to ROAR when the VWF-Selective units are hyperactivated by 3.25 times.** Sentences collapse with random numbers, irrelevant fragments, or broken logic, confirming that hyperactivating VWF-selective units does not replicate the patterns of impairment observed in humans.

| Model input | Model output |
|---|---|
| `find(real)` | The word "find" cannot be found in the image, according see me more words 123 6th of March 3, 2027, 5:42: :00—05 no-13 2016 12:00 am I am very well-/.c_0_2020_160_50_00000_132 5028_0_27_001_0_0000000. |
| `in(real)` | The word "in" is actually 1 letters. To get the maximum number of letters, you would have to look at the middle of the word. The text in the middle of this, we hope, can be as much as a as the s. The machine can be working part of its body (p) or a part.pellp the verb 'to' in[Ref#p=truly/500 word no string of fine, in which case it becomes a verb in the text of the image content. H/s=10000001/ <t>the act of |
| `never(real)` | The word "neve" completing the word 'never' is 5 characters long and 100000000 times as long as the word 'I never, by it welcome back after the 'cat' 'never'} |
| `telescoped(real)` | The word "telescopically" is a word only one time. The word "tele-scoped" can be seen in a different order. How would you support this statement with less than 10000 words at a website with a word count factoring the perfect total of each 4 and 2 before it. . 4,500 E-17. The last time with a similar tag and its own 2000.3999999999993 times [2000.1804.25000.8 |
| `everyone(real)` | The word "everyone" is not a real word. It was just a fun thing to think about. If a word is a real word and it's a word we've all seen before and the "to" means that we don't have to right now but it's a real word because it can be seen and not for other people to be used, they can't be naturally occurring as it if even where none of the time. It's then the word "eves" can be a word with one or two syllate, e.g., "be" can be used by the "best" and "you" (2 |

