# OpenReview forum: "Inducing Dyslexia in Vision Language Models"
_ICLR.cc/2026/Conference — ICLR 2026 Poster_

### Official Review · Reviewer_aZ8H · 2025-10-15

**Soundness:** 3
**Presentation:** 3
**Contribution:** 2
**Rating:** 4
**Confidence:** 4

**Summary:**

This papers studies dyslexia and the supposed role of a specific brain area, VWFA, in this disorder, corresponding to impairments in language decoding but not in normal intelligence. To do so, the authors consider a neuronal model, the Vision Language Model, learning associations between words and their visual form. The approach is in three steps: localization, ablation, testing. Neurons with the same typical responses avec the VWFA are localized and the performance of the model with or without the ablation of these neurons are measured. Interestingly, the same deficits but also the same preservation of performances are observed.

**Strengths:**

The presentation is clear, the state of the art is well presented. Applications to dyslexia are not so frequent. Results can indeed evoke a certain analogy between the artificial and natural cases.

**Weaknesses:**

Nevertheless, the approach is somewhat too superficial and does not discuss enough which conclusions can or cannot be drawn for these observations. Correlation does not mean causality.  In addition, the discussion makes perhaps a too rapid generalization by suggesting that this approach could be extended to other pathologies, including schizophrenia for example.

**Questions:**

Could you develop more the limitations of your approach, what it suggests and which kind of conclusion must be considered with caution or associated with more experiments.

---

> ### Author Response · Authors · 2025-11-24
> **Response to Reviewer aZ8H**
>
> Thank you very much for your thoughtful and detailed feedback. We sincerely appreciate your careful reading of our work.
> 1. Regarding the concern about “correlation versus causation,” we would like to clarify that in our framework, the ablation directly induces the behavioral changes we measure. In other words, we causally perturb the model units and observe the resulting behavior. Because our setup explicitly manipulates specific units and measures the downstream effect on performance, it very much provides a causal link rather than a mere correlation (see Figures 3, 4, 6, 7).
>
>
> 2. Concerning the point on generalization, our intention was to suggest that the overall framework could potentially be adapted to study other pathologies, not that our current findings already generalize beyond dyslexia. We have revised the manuscript to soften this phrasing and ensure it cannot be interpreted as an overclaim (page 10, “Broader Applicability“).
>
>
> 3. In response to the reviewer’s question about limitations, a key improvement in the revised manuscript is the inclusion of neural data and neural alignment analyses (Fig. 5, page 7) as well as generalizing our model to predict performance on dyslexia-friendly fonts (Fig. 7, page 10). These new results provide stronger evidence that the localized subnetwork in the model aligns with human VWFA activity, addressing a limitation of the original submission.
>
>
> Thank you again for your review. We kindly ask the reviewer to consider raising their score and to take into account the new results added to improve the paper, such as the neural alignment analysis and the model’s predictions regarding dyslexia-friendly fonts.

---

### Official Review · Reviewer_qoES · 2025-10-29

**Soundness:** 2
**Presentation:** 3
**Contribution:** 2
**Rating:** 2
**Confidence:** 3

**Summary:**

This paper proposes a paradigm to localize and lesion VWF-selective units in vision-language models by exploring different ablation protocols, different ways of choosing model units, and benchmarking against different clinical assessments to qualitatively capture core reading-specific deficits while preserving broader cognitive functions--which are key hallmarks of dyslexia.

**Strengths:**

The paper does a mostly thorough job of systematically introducing the research question and explaining the various cognitive assessments used to benchmark the models, and it is easy to follow. Developing mechanistic computational models of brain function that represent a rich language of hypotheses that can narrow the scope of experiments to perform ex-silico and observe trends that might be difficult in humans is an important undertaking, and I appreciate the authors' efforts in doing so.

**Weaknesses:**

(1) While the paper recapitulates findings from the dyslexia literature on a qualitative basis--such as the selective impairments in reading tasks while comprehension capabilities remain unaffected--I am not entirely convinced if that is enough to validate the framework. I know that there is really less data on brain activity or assessment scores recorded longitudinally from dyslexic individuals, but a subsequent missing quantitative comparison between the models and that data feels like a test whose absence quite strongly undermines my ability to validate the framework. (2) Additionally, could the authors point out how different their work is from the work by AlKhamissi and colleagues (2025)? Is it simply the case that their approach of functional localization is being applied to the visual word form area here? (3) Both vision and language models have been validated, to a large extent, as decent models of the corresponding sensory systems they model due to being able to use model units to explain a large amount of variance in sensory neural responses under different transform classes. A discussion on the same for vision language models which are being used here seems like a pre-requisite before even considering them to analyze brain behavior under some form of dysfunction. For example, do the representation similarity matrices from the brain and these models using some stimulus set show a large degree of convergence? (4) I appreciate the fact that the authors try a variety of different ablation strategies. However, I am curious to know why the authors think that completely ablating model units (effectively, instant cell death) is the computational principle that matches how, say, focal lesions occur in the brain. In the conclusion, the authors state their simulation is "biologically inspired",  but the ablation strategy does not seem to reveal that. (5) The framework, as it is currently set up, misses some nuance associated with modeling brain dysfunction. Dysfunction such as from focal lesions spark a cascade of ischemic processes such as the expansion of the ischemic insult into the surrounding penumbra, plasticity mechanisms, behavioral attention changes, take into account the topographic layout on a systems level, etc. While I don't consider it necessary for the paper to have recapitulated observations around all of these, at least an attempt to think about these as they pertain to dysfunction modeling seems important, and is missing.

**Questions:**

Please see the weaknesses section above.

---

> ### Author Response · Authors · 2025-11-24
> **Response to Reviewer qoES - Part 1**
>
> Thank you for your thoughtful and detailed feedback. We greatly appreciate your careful reading of our work and the constructive points you raised. We address each of your concerns below.
>
> 1. Quantitative validation and neural alignment (related to Weakness 1):
> Regarding the concern that the paper largely recapitulates dyslexia findings qualitatively and that the absence of quantitative comparison with neural data limits the validation of our framework. In response, we have added a new neural alignment analysis to the Results section (Fig. 5, page 7). This analysis demonstrates that the VWF-selective units we identify in the model show strong correspondence with human VWFA responses. These results provide additional quantitative support for the biological plausibility of our findings, helping to validate the framework beyond qualitative behavioral patterns.
>
>
> 2. Novelty relative to AlKhamissi et al. (2025) (related to Weakness 2):
> Regarding the question of how our work differs from AlKhamissi et al. (2025), we would like to clarify that our approach goes beyond applying functional localization to the VWFA by specifically modeling a neurodevelopmental disorder, dyslexia, within a vision-language model. Unlike prior work, we include multiple control conditions to assess other cognitive skills, demonstrating the behavioral specificity of VWFA-selective unit ablations. Importantly, we now provide quantitative neural alignment analyses showing that the VWF-selective units correspond closely with human VWFA responses (Fig. 5, page 7), providing mechanistic validation beyond behavioral patterns alone. Additionally, the inclusion of font prediction analyses (Fig. 7, page 10) demonstrates that the ablated model captures dyslexia-relevant sensitivities to reading-friendly and challenging fonts, something not addressed in previous studies. Together, these additions show that our framework offers both neural and behavioral specificity that goes well beyond what prior work could achieve.
>
>
>
> 3. Alignment of vision-language models with brain representations (related to Weakness 3):
> About the point that it is important to justify using VLMs for studying brain dysfunction, given prior validation of vision and language models. As noted in the manuscript (in the Related Work section on models of brain function, page 3), prior studies have shown that both vision and language models capture a substantial portion of the variance in human sensory neural responses. This motivates the use of vision-language models as a substrate for studying VWFA dysfunction, as they provide meaningful representational correspondence with human neural data. The added neural alignment results further support this justification (Fig. 5, page 7).

---

> > ### Author Response · Authors · 2025-11-24
> > **Response to Reviewer qoES - Part 2**
> >
> > 4. Biological realism of ablation strategy (related to Weakness 4):
> > Thank you for expressing the concern that completely ablating model units may not reflect biological processes such as focal lesions. We agree that ablation in models is a simplification and represents an idealized form of “instant cell death”. It is worth noting that since we only ablate a small fraction of VWF-selective units, the overall activity is slightly reduced which might correspond to a regional hypoactivation, but this is indeed an area of future research. We characterize the simulation as biologically inspired because it targets units with functional specificity similar to the VWFA in humans, allowing us to observe selective behavioral deficits analogous to dyslexia. While this does not capture all aspects of lesion dynamics, it provides a mechanistic, causal probe of the simulated internal representations.
> >
> >
> >
> > 5. Modeling nuance of brain dysfunction (related to Weakness 5)
> > Finally, you pointed out that our current framework does not capture cascades of ischemic processes, plasticity, attention shifts, or systems-level topography. We agree that these are important considerations in modeling biological dysfunction in more detail, but also note that models by definition are reductionist. Since this work is a first step toward mechanistic modeling of selective deficits rather than a full simulation of all aspects of lesion-induced dysfunction, we believe it is worth sharing the already promising results and then further refining biophysical properties. While modeling all of these is beyond the scope of the current study, future work could incorporate these processes.
> >
> >
> > Thank you again for your review. We kindly ask the reviewer to consider raising their score and to take into account the new results added to improve the paper, including the neural alignment analysis addressing the correspondence of VWFA-selective units with human neural data, as well as the model’s predictions regarding dyslexia-friendly fonts. Together with the clarifications on novelty relative to AlKhamissi et al. (2025), the justification of vision-language models as tools for studying brain function, and the discussion of the limitations of our ablation strategy, we hope these improvements strengthen the manuscript and address the concerns raised.

---

### Official Review · Reviewer_RhN7 · 2025-10-30

**Soundness:** 3
**Presentation:** 4
**Contribution:** 3
**Rating:** 6
**Confidence:** 4

**Summary:**

This paper proposes a framework for simulating dyslexia in large-scale vision-language models (VLMs) by identifying and ablating visual-word-form-selective (VWF) units, inspired by the functional role of the human visual word form area (VWFA) in reading. Using cognitive neuroscience benchmarks and targeted interventions in VLMs, the study shows that ablation of selected units impairs reading and phonological performance on tasks such as ROAR, while generally sparing visual reasoning and sentence comprehension. These findings are mostly consistent with human experiments, suggesting the potential of the proposed framework as a tool for modeling brain disorders.

**Strengths:**

1. The paper is clearly written and easy to follow. The figures are well-designed and effectively support the reader’s understanding of the main ideas.
2. Using VLMs for modeling brain disorders is a novel and interesting idea. To my knowledge, no previous studies have explored this direction.
3. The cognitive benchmarks used in this study are carefully chosen and serve well-defined purposes. The adaptation of these human tests to VLMs is clearly described and supported by sound reasoning.
4. The inclusion of multiple base models and control conditions (random ablations, model layers, etc) strengthens the study’s rigor, making the results convincing.

**Weaknesses:**

1. The core mechanisms in this paper, such as the use of VLMs and the ablation analyses, have been explored in previous neuroscience studies [1]. The authors are encouraged to cite and discuss relevant works to better situate their contributions within the existing literature.
2. The paper lacks details about the prompts used in the cognitive experiments. This is a crucial omission, as the performance and interpretability of VLMs/LLMs are known to depend heavily on prompt design. The authors should include detailed descriptions or examples of the prompts to enhance transparency and reproducibility.
3. The paper relies primarily on textual descriptions and lacks a clear mathematical formalization of the proposed framework. Incorporating explicit mathematical statements in the methodology section would improve both the scientific rigor and the reproducibility of the work.
4. It's appreciated that the authors have provided an anonymous link their source code, but the files in the repository cannot be accessed.

[1] Du, Changde, et al. "Human-like object concept representations emerge naturally in multimodal large language models." Nature Machine Intelligence (2025): 1-16.

**Questions:**

1. In Figure 3a, RAVEN accuracy increases notably as the mask size increases from 0 to 6.08%, which seems counterintuitive. Have the authors examined this phenomenon or provided an explanation for this unexpected trend?
2. In Appendix Figure 6b, PixTral exhibits substantial variation, particularly on the Raven benchmark. Could the authors clarify the cause of this high variability and whether it reflects instability in the model or differences in experimental setup?
3. It would be helpful if the authors could specify the computational resources required to conduct the experiments. This information is important for assessing the scalability and reproducibility of the proposed approach.

---

> ### Author Response · Authors · 2025-11-24
> **Response to Reviewer RhN7**
>
> Thank you very much for your thoughtful and detailed review. We greatly appreciate your careful reading of our work and the constructive points you raised. We address each of your concerns below.
> 1. Core mechanisms explored in prior work (Du et al., 2025):
> We thank you for pointing out that the use of VLMs and ablation analyses has been explored in previous studies and have now included Du et al. (2025) in the Related Work section to situate our contributions more clearly (page 3). Importantly, the key novelty of our work remains as we specifically investigate how these mechanisms relate to dyslexia by localizing VWF-selective units, performing targeted ablations, and assessing behavioral specificity relative to other cognitive skills.
>
>
> 2. Prompts for cognitive experiments:
> We agree that providing full prompt details is important for transparency and reproducibility. We have now added complete descriptions and examples of all prompts used in the cognitive experiments to the Appendix (page 15, A.3).
>
>
> 3. Mathematical formalization of the framework:
> We appreciate the suggestion, but the analyses used are relatively simple and mathematically defined in the referenced papers (student’s t-tests and Pearson correlations). We therefore decided not to include a formal mathematical description in the manuscript, as it would add little beyond what is already described in text and further specified in our open-source code release. If there is a particular aspect that would be particularly worth specifying further, please do let us know.
>
>
> 4. Source code accessibility:
> We have double-checked the repository and confirmed on multiple machines that all files are accessible. If the reviewer or others encounter issues, we are happy to provide assistance to ensure full access.
>
>
> 5. RAVEN accuracy increase (Figure 3a):
> Regarding the observation of an increase in RAVEN accuracy, this phenomenon mirrors observations in human dyslexics, who sometimes exhibit enhanced performance on nonverbal IQ or spatial reasoning tasks (Lam & Tong, 2021; von Károlyi et al., 2003; Stanovich, 2005; Tanaka et al., 2011). Consistent with this, we observe improved performance in the dyslexic model on our other nonverbal benchmark, KEMPLER (Figure 4a). We plan to investigate this phenomenon further in future work, including its potential implications for improving machine learning performance by targeted ablations of non-selective units, and have incorporated this into the Discussion (page 9, “Discussion”).
>
>
> 6. High variability of PixTral on RAVEN (Appendix Figure 8b):
> We agree that PixTral 12B (Agrawal et al., 2024) shows substantial variation on IQ-style benchmarks. Small models naturally perform poorly on such tasks (Abdelkarim et al., 2025), and this inherent limitation leads to higher variability in scores. Importantly, all runs use the same experimental setup, so these fluctuations reflect the model’s capacity rather than instability or experimental differences.
>
>
> 7. Computational resources:
> All experiments were run on two NVIDIA A100 GPUs with 80GB memory each, and introducing ablations via forward hooks adds negligible computational overhead. We specified this in the manuscript (page 17, A.5).
>
>
> 8. Neural alignment and font prediction (new analyses):
> We have added new results demonstrating that VWF-selective units show stronger alignment with human VWFA responses than random units, providing quantitative neural validation (Fig. 5, page 7). Additionally, the ablated model mirrors dyslexic readers’ sensitivity to dyslexia-friendly and challenging fonts (Fig. 7, page 10), capturing both neural and behavioral relevance.
>
>
>
> Thank you again for your review. We kindly ask the reviewer to consider raising their score and to take into account the new results, including the full prompt descriptions, neural alignment analyses, and font prediction experiments, all of which have been added to improve the paper.

---

### Official Review · Reviewer_63KZ · 2025-10-31

**Soundness:** 3
**Presentation:** 4
**Contribution:** 3
**Rating:** 4
**Confidence:** 4

**Summary:**

The paper presents a computational approach for simulating dyslexia within VLMs by identifying and ablating artificial analogues of the human visual word form area (VWFA). The authors use neuroscience-inspired functional localization to pinpoint visual-word-form-selective units inside transformer architectures and perform targeted ablation to mimic the neural disruptions observed in dyslexic individuals. Ablation leads to substantial reading deficits, measured by clinical benchmarks, while leaving general visual reasoning and sentence comprehension unaffected, thus reproducing the dissociation observed in human dyslexia.

Critically, the computational lesion predominantly impairs phonological processing with minimal orthographic deficits, paralleling key aspects of developmental dyslexia in the clinic. The framework is benchmarked across multiple VLMs and analyzed for specificity with rigorous controls, and is offered as a platform for causally testing mechanisms of neurodevelopmental disorders using artificial neural networks.

**Strengths:**

- The work is original in proposing unit-level functional localization and lesioning inside VLMs to simulate neurobiological disorders. Unlike previous works, the method operates at the level of hypothesized neural substrates and performs causal manipulations aligned with neuroscience conventions rather than coarse-grained connectivity perturbations or behavioral proxies.​

- The experimental design is precise: identification of VWF-selective units via well-established localizer paradigms from human neuroimaging is directly transposed to transformer architectures, and quantitative assessment is conducted with adapted human clinical tasks.​

- Control experiments using random unit ablation convincingly show that only lesioning functionally localized word-selective units produces the desired dissociation, with random ablation causing global performance degradation, establishing the necessity of neuro-informed unit targeting.​

- The work addresses not just cognitive outcomes but fidelity to documented human error types; response analyses detail patterns like misclassification, contextual over-interpretation, ambiguous hedging, and corrupted outputs, directly paralleling error modes seen in clinical dyslexia assessments.​

- The quantitative analysis is comprehensive, including hyperparameter sweeps for mask size, scaling factor variations, and layer-type discrimination, allowing for mechanistic insights into where reading specialization resides within transformer blocks.​

**Weaknesses:**

- The identification of VWF-selective units depends on activation statistics from model internals in response to image categories, but does not incorporate human brain recordings! Without direct mapping between artificial and biological VWFA units, the correspondence remains functional rather than structural or neural, potentially limiting claims about mechanisms and transferability.​

- The clinical profiles of dyslexia captured by the ablated models are more purely phonological than in human subjects, who often display mixed phonological and orthographic impairments. The model’s orthographic capacities are notably resistant to ablation, and the observed deficit distribution is less nuanced than seen in patients, raising questions as to whether this reflects architectural constraints or a misalignment in the simulated lesion procedure. The authors acknowledge this, but propose no practical avenues for resolving it within the current model scope?

- The experimental controls, while thorough, operate entirely within model space. There is no demonstration that the artificial VWFA units share structural, developmental, or topological features with their biological counterpart. For example, are the localized units spatially clustered within the transformer, and do their connectivity patterns resemble cortical circuits?

- Evaluation benchmarks are based on image-word discrimination, general nonverbal reasoning, and sentence comprehension tasks, all adapted from clinical settings. Although valid for cross-domain comparison, the complexity of the clinical phenotype is reduced by focusing only on limited aspects (speed of reading is omitted; real-life reading comprehension and development trajectory are not simulated). Dyslexia is developmental, but the ablation is static and does not reflect learning dynamics.

**Questions:**

- Can the identification of VWF-selective units be validated by alignment with neural data, for example, through representational similarity analysis with fMRI or MEG signals from human readers? This would strengthen claims of biological realism.

- The phonological deficit induced by ablation is robust, but the orthographic deficit is comparatively minor. Can the lesioning approach be modified to target orthographic processing more effectively, and what does this suggest about the separability of these mechanisms in transformer architectures?

- Are the identified VWF-selective units spatially or functionally clustered within the model, and do they possess unique connectivity profiles? Are there architectural analogues to known properties of the VWFA (position, connectivity) in the cortex that can be drawn from transformer weights or activation maps?

- The current approach is limited to the ablation (silencing) of units. Have the authors considered developmentally inspired interventions, such as training models under restricted input, gradual lesioning, or biased gradient propagation, to better capture the emergence and remediation of dyslexia-like deficits?

- Benchmark choices are justified, but the absence of processing speed (a key diagnostic marker in dyslexia) is notable. Can future work incorporate time-dependent metrics?

- Minor: Visualizations (activation maps via GradCAM, ablated unit distributions) could benefit from more direct comparison to human neuroimaging data or developmental profiles.

---

> ### Author Response · Authors · 2025-11-24
> **Response to Reviewer 63KZ - Part 1**
>
> Thank you very much for your careful reading of our work and for the thoughtful comments. We appreciate the opportunity to clarify and expand on several points raised in your review. Below, we address each concern in turn.
>
>
>
>
> 1. Neural alignment validation:
> Regarding the observation that the identification of VWF-selective units depends only on model activations and not on human neural recordings: we would have loved to compare model activity to brain data right away, but until recently such data was not publicly available. Thanks to a recent release by the Kanwisher lab (Marvi et al., 2025) however, we have now added a neural alignment analysis in the Results section. Using a standard encoding-model framework, we show that the model’s VWF-selective units indeed predict human VWFA responses significantly better than random subsets of units (Fig. 5, page 7). This provides functional validation that these units capture human-relevant neural representations.
>
>
> 2. Predominantly phonological dyslexia:
> Regarding the concern that the model produces mostly phonological deficits while clinical dyslexia often shows both phonological and orthographic impairments: we agree that human dyslexia frequently involves mixed deficits on average. However, isolating VWFA-specific deficits in humans is widely recognized as challenging, due to frequent co-occurrence with broader linguistic or attentional disorders. The human datasets we examined do not contain a population with a clean, selective VWFA impairment. Our model allows us to isolate this deficit precisely: ablating only the VWFA-analogous units produces a predominantly phonological profile. This pattern is consistent with the dominant view in neuroscience and clinical literature, which characterizes dyslexia linked to VWFA or left-hemispheric reading-circuit dysfunction as primarily phonological in nature (Peterson & Pennington, 2012). Thus, the model’s phonology-focused impairment does not contradict clinical findings; rather, it reflects the expected outcome of a selective VWFA-linked disruption. One key advantage of our approach is the ability to test mechanistic hypotheses under controlled conditions that are not feasible in human research.
>
>
>
> 3. Structural or developmental homology:
> The absence of structural homology does not limit the framework’s practical usefulness. Because the ablated (“dyslexic”) model reliably reproduces behavioral patterns observed in dyslexic readers, it can support intervention-oriented applications. To illustrate this point, we ran a new analysis where we evaluated the model on fonts known to be difficult or dyslexia-friendly – indeed, the ablated model mirrors human sensitivity: it performs worse or remains unchanged on challenging fonts, and improves or remains stable on recommended dyslexia-friendly fonts, whereas the intact model shows no such effects (Fig. 7, page 10).
> This opens a possibility that is unique to a computational framework: we can now design or optimize new fonts automatically using the model itself, by searching for visual configurations that maximize performance in the dyslexic model while preserving readability in the intact model. We have discussed this point in the manuscript (page 9 “Predicting Dyslexia-Friendly Fonts”  and 10 “Discussion”) to clarify that functional alignment alone can generate actionable insights and support the development of dyslexia-friendly interventions.
>
>
> 4. Developmentally inspired interventions:
> Regarding the possibility of training models under restricted input, gradual lesioning, or biased gradient propagation: we agree that these approaches could provide complementary insights into developmental emergence and remediation of dyslexia. Our current approach, ablating VWFA-analogous units, reflects deficits observed in developmental dyslexia (Mitchell et al., 2025). Future work could explore gradual or biased manipulations to approximate developmental dynamics.

---

> ### Author Response · Authors · 2025-11-24
> **Response to Reviewer 63KZ - Part 2**
>
> 5. Time-dependent metrics:
> Regarding the processing speed or time-dependent benchmarks: the current models do not explicitly encode time, limiting their ability to capture speed-dependent reading measures. However, future work could leverage intermediate outputs (e.g., logit evolution across decoding steps) to approximate temporal processing or reading speed in dyslexia. We have added this interesting idea as a future direction in the manuscript (page 10, “Discussion”).
>
>
> 6. Computational resources:
> Regarding computational resources: all experiments were run on two NVIDIA A100 GPUs with 80GB memory each. The ablation implementation does not introduce meaningful overhead. We specified this in the manuscript (page 17, A.5).
>
>
> 7. Additional new analyses:
> We have also added analyses demonstrating that the ablated model mirrors human dyslexic behavior in font sensitivity and that the VWF-selective units predict human VWFA responses better than random units. These results strengthen the biological and behavioral relevance of our framework and support its potential utility in guiding intervention strategies.
>
>
> Thank you again for your review. We kindly ask the reviewer to consider raising their score and to take into account the new results added to improve the paper, including the neural alignment analysis addressing the correspondence of VWFA-selective units with human neural data, as well as the model’s predictions regarding dyslexia-friendly fonts.

---

### Author Response · Authors · 2025-12-03
**General Comment**

We thank all reviewers for their thoughtful and constructive feedback. We have carefully addressed major and minor concerns, and the revised paper is substantially stronger as a result. In brief, we confirmed that the identified model units are neurally predictive of human VWFA brain recordings, and that our model behaviorally predicts which fonts are easier for individuals with dyslexia. Below we summarize the key updates and clarifications for the Area Chair’s consideration.

&nbsp;

# Neural alignment of model units and brain recordings
Several reviewers requested quantitative evidence linking the model’s VWF-selective units to the human VWFA. With the recent release of Marvi et al. (2025), we were able to add a full neural alignment analysis (Fig. 5, page 7). Using a standard encoding-model framework, we show that VWF-selective units predict human VWFA responses significantly better than randomly sampled units. This provides direct biological validation of our localized units and addresses the reviewers’ core concerns regarding mechanistic realism.

&nbsp;

# Behavioral validation via font-sensitivity experiments
To further show our ablation captures clinically meaningful aspects of dyslexia, we added new experiments evaluating the model on dyslexia-relevant fonts (pages 9–10). The dyslexic model reproduces the elevated sensitivity to visually challenging fonts that is reported in dyslexic readers, while the intact model does not. Importantly, this also serves as a proof-of-concept demonstration of how the model could be used to evaluate or design fonts that reduce visual stress.

&nbsp;

# Clearer articulation of novelty and relationship to prior work
We have strengthened the positioning of our contributions relative to AlKhamissi et al. (2025) and Du et al. (2025). Our work goes beyond prior studies by introducing a mechanistic framework for simulating a neurodevelopmental reading disorder inside a vision-language model. By identifying VWFA-analogous units with a neuroimaging-inspired localizer, causally perturbing them, and demonstrating both neural and behavioral specificity, including the two new validation analyses, our approach provides a new computational tool for understanding dyslexia. To the best of our knowledge, no prior work has modeled dyslexia (or other brain disorders) at this mechanistic level within a VLM.

&nbsp;

# Improved transparency and reproducibility
Following reviewer suggestions, we have added full prompt descriptions and examples (Appendix A.3) and clarified computational resources (Appendix A.5).

&nbsp;

# Clarified modeling choices and limitations
Finally, we expanded discussion around biological realism, lesion modeling, and developmental considerations (page 10). While computational models inevitably simplify neural processes, our approach mirrors established VWFA-linked deficits and offers a uniquely controlled environment for testing causal hypotheses that are experimentally not easily accessible in human participants.

&nbsp;


# Overall contribution and impact
With the addition of neural alignment results and behavioral font-sensitivity experiments, the paper now brings together convergent neural and behavioral evidence that the model’s impairments authentically parallel dyslexia. We believe the revised submission represents a well-validated, and novel step toward using large-scale models to study and potentially assist with neurodevelopmental reading disorders such as dyslexia.

&nbsp;

We kindly ask the Area Chair to consider these substantial improvements and clarifications in the final decision process. For further details on minor improvements and responses to specific reviewer comments, we invite the Area Chair to consult the responses provided below.

---

### Meta-Review · Area_Chair_abMt · 2025-12-18

**Summary:**

**Summary:**

This article provides a method to induce dyslexia (a human disorder that impairs or reduces reading abilities) in vision-language models by ablating units that are selective towards visual word form. As a result, several properties of human dyslexia are modeled by this procedure, such as phonological deficits without affecting orthographic processing and dissociating other measures (e.g. visual reasoning). Together, the approach provides a computational model (based on an open-source VLM) of dyslexia that could serve as a starting point for future investigations in that area, towards the author’s vision of a “digital twin” that allow for experiments which cannot be conducted in humans since they rely on invasive methods.

**Recommendation:**

The article advances our understanding of modeling dyslexia with VLMs, and the authors have addressed limitations & suggestions during the discussion period. For these reasons, acceptance is recommended.

**Reviewer Concerns:**

Discussion / Rebuttal summary:
Initial review scores were mostly borderline (6,2,4,4), with the lowest score (2) also being the lowest confidence score.

- Review ``63KZ`` is very likely completely AI-generated (according to an AI detection tool) and contains nonsensical points (e.g., advocating that "processing speed" between VLMs and humans be compared: while humans indeed process different words at different speeds, there's no such processing speed difference for different text tokens in language models). For this reason, the review is being ignored by the AC.

- The suggestions of reviewer ``RhN7`` have been largely addressed by the authors except for a mathematical formalization, which is beyond the scope of the (empirical) article.

- Review ``qoES`` raises a number of different questions; the authors provided a detailed response that addresses some concerns and clarifies aspects of their submission, for example by adding Figure 5 as a quantitative figure which is based on a recommendation / concern of the reviewer.

- Review ``aZ8H`` is on the short end; actionable suggestions & questions were addressed by the authors who also cleared up a misunderstanding about correlation vs. causation, confirming that their ablation approach is indeed a causal (not correlational) approach. The reviewer concern that “the approach is somewhat too superficial” is too vague / unspecific to be actionable; the authors made a pass over their discussion to avoid overclaiming.

**Reviewer Scores:**

See above - since the authors successfully addressed important points raised in the initial reviews, the post-rebuttal scores would consequently have been higher, advocating for acceptance.

---

### Decision · Program_Chairs · 2026-01-26

Accept (Poster)